# DIFFUSION ON SYNTAX TREES FOR PROGRAM SYNTHESIS

**Shreyas Kapur, Erik Jenner & Stuart Russell**
University of California, Berkeley
{srkp,jenner,russell}@berkeley.edu

## ABSTRACT

Large language models generate code one token at a time. Their autoregressive generation process lacks the feedback of observing the program's output. Training LLMs to suggest edits directly can be challenging due to the scarcity of rich edit data. To address these problems, we propose neural diffusion models that operate on syntax trees of any context-free grammar. Similar to image diffusion models, our method also inverts "noise" applied to syntax trees. Rather than generating code sequentially, we iteratively edit it while preserving syntactic validity, which makes it easy to combine this neural model with search. We apply our approach to inverse graphics tasks, where our model learns to convert images into programs that produce those images. Combined with search, our model is able to write graphics programs, see the execution result, and debug them to meet the required specifications. We additionally show how our system can write graphics programs for hand-drawn sketches. Video results can be found at https://tree-diffusion.github.io.

## 1 INTRODUCTION

Large language models (LLMs) have made remarkable progress in code generation, but their autoregressive nature presents a fundamental challenge: they generate code token by token, without access to the program's runtime output from the previously generated tokens. This makes it difficult to correct errors, as the model lacks the feedback loop of seeing the program's output and adjusting accordingly. While LLMs can be trained to suggest edits to existing code (Chakraborty et al., 2020; Zhang et al., 2022; Jin et al., 2023), acquiring sufficient training data for this task is difficult.

In this paper, we introduce a new approach to program synthesis using *neural diffusion* models that operate directly on syntax trees. Diffusion models have previously been used to great success in image generation (Ho et al., 2020; Nichol et al., 2021; Song et al., 2020). By leveraging diffusion, we let the model learn to iteratively refine programs while ensuring syntactic validity. Crucially, our approach allows the model to observe the program's output at each step, effectively enabling a debugging process.

In the spirit of systems like AlphaZero (Silver et al., 2018), the iterative nature of diffusion naturally lends itself to search-based program synthesis. By training a value model alongside our diffusion model, we can guide the denoising process toward programs that are likely to achieve the desired output. This allows us to efficiently explore the program space, making more informed decisions at each step of the generation process.

We implement our approach for inverse graphics tasks, a problem of interest in the symbolic program synthesis literature (Ellis et al., 2021; 2019). This domain is naturally suitable for our approach because small changes in the code produce semantically meaningful changes in the rendered image. We are further motivated to explore this domain because modern vision-language models (VLMs) struggle to achieve exact pixel matches on these tasks (Figure 4).

Our main contributions for this work are (a) a novel approach to program synthesis using diffusion on syntax trees and (b) an implementation of our approach for inverse graphics tasks that significantly outperforms previous methods.

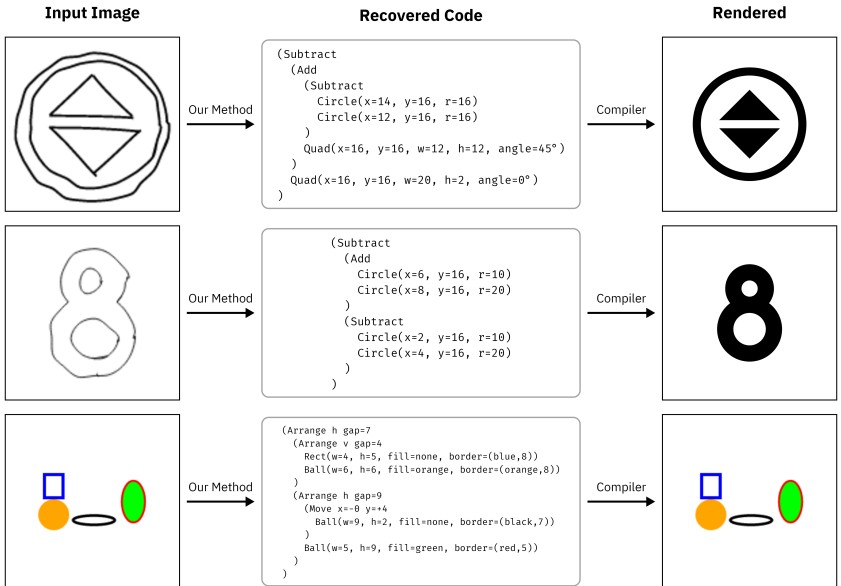

Figure 1: Examples of programs recovered by our system. The top row shows a hand-drawn sketch of an icon (left), the recovered program (middle), and the compilation of the recovered program (right). The top two rows are for the constructive solid geometry language (`CSG2D-Sketch`). The last row is an example output from our `TinySVG` environment that learns to invert hierarchical programs of shapes and colors. Video examples can be found at `https://tree-diffusion.github.io`.

## 2 BACKGROUND & RELATED WORK

**Neural program synthesis** Neural program synthesis is a prominent area of research, in which neural networks generate programs from input-output examples. Early work, such as Parisotto et al. (2016), demonstrated the feasibility of this approach. While modern language models can be directly applied to program synthesis, combining neural networks with search strategies often yields better results and guarantees. In this paradigm, the neural network guides the search process by providing proposal distributions or scoring candidate programs. Examples of such hybrid methods include Balog et al. (2016), Ellis et al. (2021), and Devlin et al. (2017). A key difference from our work is that these methods construct programs incrementally, exploring a vast space of partial programs. Our approach, in contrast, focuses on *editing* programs, allowing us to both grow programs from scratch and make corrections based on the program execution.

**Neural diffusion** Neural diffusion models, a class of generative models, have demonstrated impressive results for modeling high-dimensional data, such as images (Ho et al., 2020; Nichol et al., 2021; Song et al., 2020). A neural diffusion model takes samples from the data distribution (e.g. real-world images), incrementally corrupts the data by adding noise, and trains a neural network to incrementally remove the noise. To generate new samples, we can start with random noise and iteratively apply the neural network to denoise the input.

**Diffusion for discrete data** Recent work extends diffusion to discrete and structured data like graphs (Vignac et al., 2022), with applications in areas such as molecule design (Hoogeboom et al., 2022; Schneuing et al., 2022; Corso et al., 2022). Notably, Lou et al. (2023) proposed a discrete diffusion model using a novel score-matching objective for language modeling. Another promising line of work for generative modeling on structured data is generative flow networks (GFlowNets) (Bengio et al., 2023), where neural models construct structured data one atom at a time.

**Diffusion for code generation** Singh et al. (2023) use a diffusion model for code generation. However, their approach is to first embed text into a continuous latent space, train a *continuous*

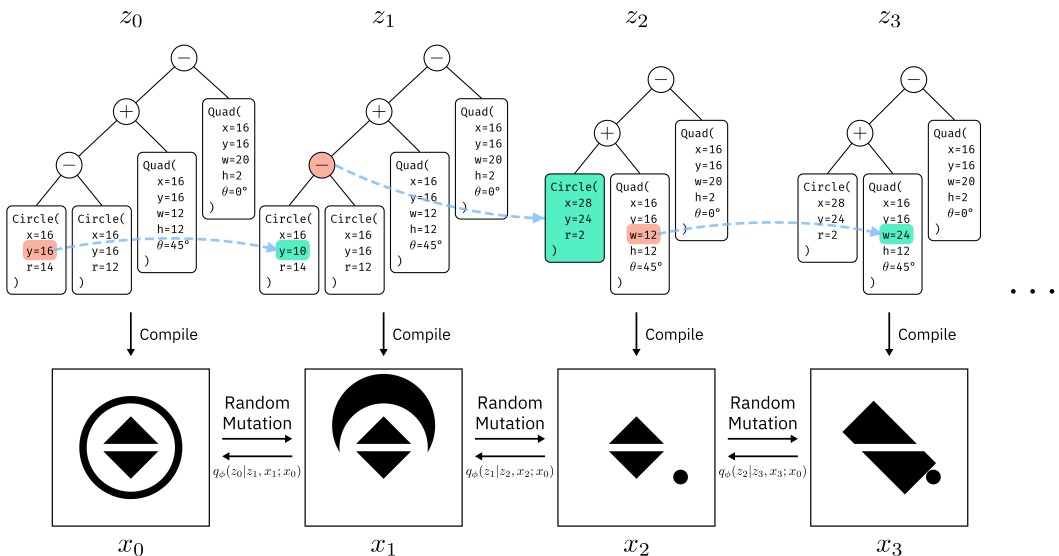

Figure 2: An overview of our method. Analogously to adding noise in image diffusion, we randomly make small mutations to the syntax trees of programs. We then train a conditional neural model to invert these small mutations. In the above example, we operate in a domain-specific language (DSL) for creating 2D graphics using a constructive solid geometry language. The leftmost panel ($z_0$) shows the target image (bottom) alongside its program as a syntax tree (top). The $y$ value of the circle gets mutated from 16 to 10 in the second panel, making the black circle "jump" a little higher. Between $z_1$ and $z_2$, we see that we can mutate the Subtract ($-$) node to a Circle node, effectively deleting it.

diffusion model on that space, and then unembed at the end. This means that intermediate stages of the latent representation are not trained to correspond to actual code. The embedding tokens latch to the nearest embeddings during the last few steps.

Direct code editing using neural models has also been explored. Chakraborty et al. (2020) use a graph neural network for code editing, trained on a dataset of real-world code patches. Similarly, Zhang et al. (2022) train a language model to edit code by modifying or inserting [MASK] tokens or deleting existing tokens. They further fine-tune their model on real-world comments and patches. Unlike these methods, our approach avoids the need for extensive code edit datasets and inherently guarantees syntactic validity through our pretraining task. ChainCoder (Zheng et al., 2023) used the abstract-syntax tree structure explicitly for code generation with language models, but do not use execution guidance like our approach.

**Program synthesis for inverse graphics** We are inspired by previous work by Sharma et al. (2018); Ellis et al. (2018; 2019), which also uses the CSG2D language. Sharma et al. (2018) propose a convolutional encoder and a recurrent model to go from images to programs. Ellis et al. (2019) propose a method to provide a neural model with the intermediate program execution output in a read–eval–print loop (REPL). Unlike our method, the ability to execute partial graphics programs is a key requirement for their work. Our system operates on complete programs and does not require a custom partial compiler. As mentioned in their work, their policies are also brittle. Once the policy proposes an object, it cannot undo that proposal. Hence, these systems require a large number of particles in a Sequential Monte-Carlo (SMC) sampler to make the system less brittle to mistakes.

## 3 METHOD

The main idea behind our method is to develop a form of denoising diffusion models analogous to image diffusion models for syntax trees.

Consider the example task from Ellis et al. (2019) of generating a constructive solid geometry (`CSG2D`) program from an image. In `CSG2D`, we can combine simple primitives like circles and quadrilaterals using boolean operations like addition and subtraction to create more complex shapes, with the context-free grammar (CFG),

$$S \rightarrow S + S \,|\, S - S \,|\, \texttt{Circle}_{x,y}^{r} \,|\, \texttt{Quad}_{x,y,\theta}^{w,h}.$$

In Figure 2, $z_0$ is our *target program*, and $x_0$ is the rendered version of $z_0$. Our task is to invert $x_0$ to recover $z_0$. Our noising process randomly mutates y=16 to y=10. It then mutates the whole $\ominus$ sub-tree with two shapes with a new sub-tree with just one shape. Conditioned on the image $x_0$, and starting at $z_3, x_3$, we would like to train a neural network to reverse this noising process to get to $z_0$.

In the following sections, we will first describe how "noise" is added to syntax trees. Then, we will detail how we train a neural network to reverse this noise. Finally, we will describe how we use this neural network for search.

## 3.1 SAMPLING SMALL MUTATIONS

Let $z_t$ be a program at time $t$. Let $p_{\mathcal{N}}(z_{t+1}|z_t)$ be the distribution over randomly mutating program $z_t$ to get $z_{t+1}$. We want $p_{\mathcal{N}}$ mutations to be: (1) small and (2) produce syntactically valid $z_{t+1}$'s.

To this end, we turn to the rich computer security literature on grammar-based fuzzing (Zeller et al., 2023; Godefroid et al., 2008; Srivastava & Payer, 2021; Wang et al., 2019). To ensure the mutations are small, we first define a function $\sigma(z)$ that gives us the "size" of program $z$. For all our experiments, we define a set of terminals in our CFG to be *primitives*. As an example, the primitives in our `CSG2D` language are $\{\texttt{Quad}, \texttt{Circle}\}$. In that language, we use $\sigma(z) = \sigma_{\text{primitive}}(z)$, which counts the number of primitives. Other generic options for $\sigma(z)$ could be the depth, number of nodes, etc.

We then follow Luke (2000) and Zeller et al. (2023) to randomly sample programs from our CFG under exact constraints, $\sigma_{\min} < \sigma(z) \leq \sigma_{\max}$. We call this function `ConstrainedSample`$(\sigma_{\min}, \sigma_{\max})$. Setting a small value for $\sigma_{\max}$ allows us to sample *small* programs randomly. We set $\sigma_{\max} = \sigma_{\text{small}}$ when generating small mutations.

To mutate a given program $z$, we first generate a set of candidate nodes in its tree under some $\sigma_{\text{small}}$,

$$\mathcal{C} = \{n \in \texttt{SyntaxTree}(z) \,|\, \sigma(n) \leq \sigma_{\text{small}}\}.$$

Then, we uniformly sample a mutation node from this set,

$$m \sim \text{Uniform}[\mathcal{C}].$$

Since we have access to the full syntax tree and the CFG, we know which production rule produced $m$, and can thus ensure syntactically valid mutations. For example, if $m$ were a number, we know to replace it with a number. If $m$ were a general subexpression, we know we can replace it with any general subexpression. Therefore, we sample $m'$, which is $m$'s replacement, as,

$$m' \sim \texttt{ConstrainedSample}(\texttt{ProductionRule}(m), \sigma_{\text{small}}).$$

## 3.2 POLICY

### 3.2.1 FORWARD PROCESS

We cast the program synthesis problem as an inference problem. Let $p(x|z)$ be our observation model, where $x$ can be any kind of observation. For example, we will later use images $x$ produced by our program, but $x$ could also be an execution trace, a version of the program compiled to bytecode, or simply a syntactic property. Our task is to invert this observation model, i.e. produce a program $z$ given some observation $x$.

We first take some program $z_0$, either from a dataset, $\mathcal{D} = \{z^0, z^1, \ldots\}$, or in our case, a randomly sampled program from our CFG. We sample $z_0$'s such that $\sigma(z_0) \leq \sigma_{\max}$. We then add noise to $z_0$ for $s \sim \text{Uniform}[1, s_{\max}]$, steps, where $s_{\max}$ is a hyper-parameter, using,

$$z_{t+1} \sim p_{\mathcal{N}}(z_{t+1}|z_t).$$

We then train a conditional neural network that models the distribution,

$$q_{\phi}(z_{t-1}|z_t, x_t; x_0),$$

where $\phi$ are the parameters of the neural network, $z_t$ is the current program, $x_t$ is the current output of the program, and $x_0$ is the target output we are solving for.

### 3.2.2 REVERSE MUTATION PATHS

Since we have access to the ground-truth mutations, we can generate targets to train a neural network by simply reversing the sampled trajectory through the forward process Markov-Chain, $z_0 \rightarrow z_1 \rightarrow \ldots \rightarrow z_T$. At first glance, this may seem a reasonable choice. However, training to simply invert the last mutation can potentially create a much noisier signal for the neural network.

Consider the case where, within a much larger syntax tree, a color was mutated as,

$$\text{Red} \rightarrow \text{Blue} \rightarrow \text{Green}.$$

The color in our target image, $x_0$, is Red, while the color in our mutated image, $x_2$, is Green. If we naively teach the model to invert the above Markov chain, we are training the network to turn the Green to a Blue, even though we could have directly trained the network to go from Green to a Red.

Therefore, to create a better training signal, we compute an *edit path* between the target tree and the mutated tree. We use a tree edit path algorithm loosely based on the tree edit distance introduced by Pawlik & Augsten (2016; 2015). The general tree edit distance problem allows for the insertion, deletion, and replacement of any node. Unlike them, our trees can only be edited under an action space that only permits *small* mutations. For two trees, $z_A$ and $z_B$, we linearly compare the syntax structure. For changes that are already $\leq \sigma_{\text{small}}$, we add that to our mutation list. For changes that are $> \sigma_{\text{small}}$, we find the first mutation that reduces the distance between the two trees. Therefore, for any two programs, $z_A$ and $z_B$, we can compute the first step of the mutation path in $O(|z_A| + |z_B|)$ time.

## 3.3 VALUE NETWORK & SEARCH

We additionally train a value network, $v_\phi(x_A, x_B)$, which takes as input two rendered images, $x_A$ and $x_B$, and predicts the edit distance between the underlying programs that generated those images. Since we have already computed edit paths between trees during training, we have direct access to the ground-truth program edit distance for any pair of rendered images, allowing us to train this value network in a supervised manner.

Using our policy, $q_\phi(z_{t-1}|z_t, x_t; x_0)$, and our value, $v_\phi(x_{t_A}, x_{t_B})$, we can perform beam-search for a given target image, $x_0$, and a randomly initialized program $z_t$. At each iteration, we maintain a collection of nodes in our search tree with the most promising values and only expand those nodes.

## 3.4 ARCHITECTURE

Figure 3 shows an overview of our neural architecture. We use a vision-language model described by Tsimpoukelli et al. (2021) as our denoising model, $q_\phi(z_{t-1}|z_t, x_t; x_0)$. We use an off-the-shelf implementation (Wightman, 2019) of NF-ResNet-26 as our image encoder, which is a normalizer-free convolutional architecture proposed by Brock et al. (2021) to avoid test time instabilities with Batch-Norm (Wu et al., 2023). We implement a custom tokenizer, using the terminals of our CFG as tokens. The rest of the edit model is a small decoder-only transformer (Vaswani et al., 2017; Radford et al., 2019), with a total model size of $\approx 10$ million parameters.

We add two additional types of tokens: an <EDIT> token, which serves as a start-of-sentence token for the model; and <POS x> tokens, which allow the model to reference positions within its context. Given a current image, a target image, and a current tokenized program, we train this transformer model to predict the edit position and the replacement text autoregressively. While making predictions, the decoding is constrained under the grammar. We mask out the prediction logits to only include edit positions that represent nodes in the syntax tree, and only produce replacements that are syntactically valid for the selected edit position.

We set $\sigma_{\text{small}} = 2$, which means the network is only allowed to produce edits with fewer than two primitives. For training data, we sample an infinite stream of random expressions from the CFG. We choose a random number of noise steps, $s \in [1, 5]$, to produce a mutated expression. For some percentage of the examples, $\rho$, we instead sample a completely random new expression as our mutated expression. We trained for 3 days for the environments we tested on a single Nvidia A6000 GPU.

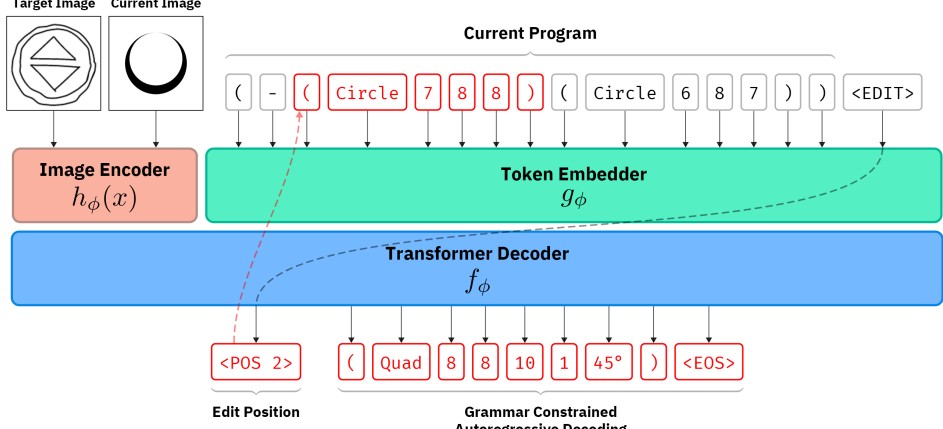

Figure 3: We train $q_\phi(z_{t-1}|z_t, x_t; x_0)$ as a decoder only vision-language transformer following Tsimpoukelli et al. (2021). We use an NF-ResNet as the image encoder, which is a normalizer-free convolutional architecture proposed by Brock et al. (2021). The image encoder encodes the current image, $x_t$, and the target images, $x_0$. The current program is tokenized according to the vocabulary in our context-free grammar. The decoder first predicts an *edit* location in the current program, and then tokens that replace what the edit location should be replaced by. We constrain the autoregressive decoding by our context-free grammar by masking only the valid token logits.

## 4 EXPERIMENTS

### 4.1 ENVIRONMENTS

We conduct experiments on four domain-specific graphics languages, with complete grammar specifications provided in Appendix B.

**CSG2D**  A 2D constructive solid geometry language where primitive shapes are added and subtracted to create more complex forms, as explored in our baseline methods (Ellis et al., 2019; Sharma et al., 2018). We also create `CSG2D-Sketch`, which has an added observation model that simulates hand-drawn sketches using the algorithm from Wood et al. (2012).

**TinySVG**  A language featuring primitive shapes with color, along with `Arrange` commands for horizontal and vertical alignment, and `Move` commands for shape offsetting. Figure 1 portrays an example program. Unlike the compositional nature of `CSG2D`, `TinySVG` is hierarchical: sub-expressions can be combined into compound objects for high-level manipulation. We also create, `Rainbow`, a simplified version of `TinySVG` without `Move` commands for ablation studies due to its reduced computational demands.

We implemented these languages using the Lark (Lark Contributors, 2014) and Iceberg (IceBerg Contributors, 2023) Python libraries, with our tree-diffusion implementation designed to be generic and adaptable to any context-free grammar and observation model.

### 4.2 BASELINES

We use two prior works, Ellis et al. (2019) and the vision-language model (VLM) suggested by Tsimpoukelli et al. (2021) in the spirit of Sharma et al. (2018) as baseline methods.

**VLM**  Sharma et al. (2018) employed a convolutional and recurrent neural network to generate program statements from an input image. For a fair comparison, we re-implemented CSGNet using the same vision-language transformer architecture (VLM) as our method, representing the modern autoregressive approach to code generation. We use rejection sampling, repeatedly generating programs until a match is found.

**REPL VLM**    Ellis et al. (2019) proposed a method to build programs one primitive at a time until all primitives have been placed. They also give a policy network access to a REPL, i.e., the ability to execute code and see outputs. Notably, this *current image* is rendered from the current *partial program*. As such, we require a custom *partial compiler*. This is straightforward for `CSG2D` since it is a compositional language. We simply render the shapes placed so far. For `TinySVG`, it is not immediately obvious how this partial compiler should be written. This is because the rendering happens bottom-up. Primitives get arranged, and those arrangements get arranged again (see Figure 1). Therefore, we only use this baseline method with `CSG2D`.

**Tree Diffusion**    We split our approach into two. The first, Tree Diffusion Search, uses the beam search method previously described, that leverages both a policy network and a value network. This allows us to explore the possible program solutions more effectively, guided by the output generated at each step of the program's execution. The second part, Tree Diffusion Rollouts, relies solely on the policy network. It repeatedly samples potential edit actions from the policy's distribution until it arrives at a program that satisfies the given specification.

**Test tasks**    For `TinySVG` we used a held-out test set of randomly generated expressions and their images. For the `CSG2D` task, we noticed that all methods were at ceiling performance on an in-distribution held-out test set. In Ellis et al. (2019), the authors created a harder test set with more objects. However, simply adding more objects in an environment like `CSG2D` resulted in simpler final scenes, since sampling a large object that subtracts a large part of the scene becomes more likely. Instead, to generate a hard test set, we filtered for images at the $95th$ percentile or more on incompressibility with the LZ4 (Collet et al., 2013; Welch, 1984) compression algorithm.

**Evaluation**    In `CSG2D`, we accepted a predicted program as matching the specification if it achieved an intersection-over-union (IoU) of 0.99 or more. In `TinySVG`, we accepted an image if 99% of the pixels were within $0.005 \approx \frac{1}{256}$. It is notably very challenging to pass this strict requirement to match almost all the pixels. In Appendix D, Figure 11, we show the complexity of these test tasks.

All methods were trained with supervised learning and were not fine-tuned with reinforcement learning. The training data was generated by using our `ConstrainedSample` function, which randomly samples programs from grammars. All methods used the grammar-based constrained decoding method described in Section 3.4, which ensured syntactically correct outputs. While testing, we measured performance based on the number of nodes the method needed to expand to complete the task. For each method, a node expansion is a call to the neural network and a rendering step to check the answer. Since all methods tested use an identical neural architecture and parameter counts, "number of nodes expanded" is a comparable metric of total amount of computation required by each of the methods.

Figure 4 shows the performance of our method compared to the baseline methods. In both the `CSG2D` and `TinySVG` environments, our tree diffusion policy rollouts significantly outperform the policies of previous methods. Our policy combined with beam search further improves performance, solving problems with fewer calls to the renderer than all other methods. Figure 6 shows successful qualitative examples of our system alongside outputs of baseline methods. We note that our system can fix smaller issues that other methods miss. Figure 7 shows some examples of recovered programs from sketches in the `CSG2D-Sketch` language, showing how the observation model does not necessarily need to be a deterministic rendering; it can also consist of stochastic hand-drawn images.

### 4.3    ABLATIONS

To understand the impact of our design decisions, we performed ablation studies on the simplified `Rainbow` environment using a smaller transformer model.

First, we examined the effect of removing the current image (no REPL) from the policy network's input. As shown in Figure 5(a), this drastically hindered performance, confirming the importance of a REPL-like interface observed by Ellis et al. (2019).

Next, we investigated the necessity of our reverse mutation path algorithm. While training on the last mutation step alone provides a valid path, it introduces noise by potentially targeting subopti-

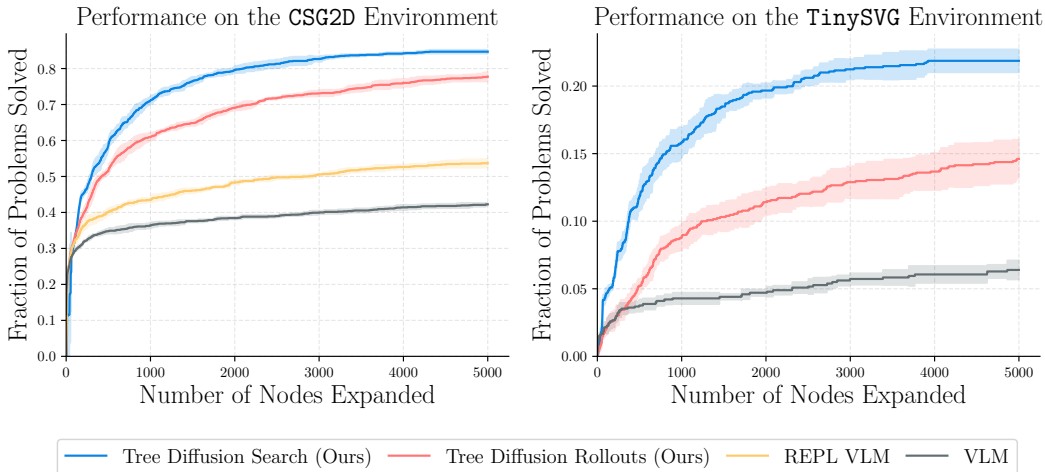

Figure 4: Performance of our approach in comparison to baseline methods in `CSG2D` and `TinySVG` languages. We give the methods $n = 256$ images from the test set and measure the number of nodes expanded to find a solution. The auto-regressive baseline was queried with rejection sampling. Our policy outperforms previous methods, and our policy combined with search helps boost performance further. All neural networks here have the same architecture and number of parameters. Error bars show standard deviation across 5 random seeds.

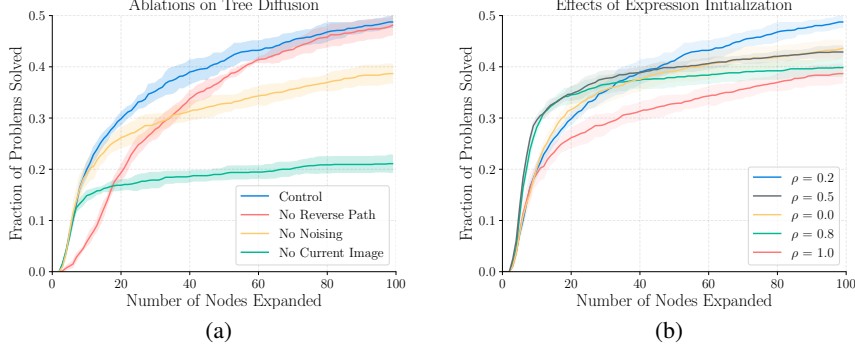

Figure 5: Effects of changing several design decisions of our system. We train smaller models on the `Rainbow` environment. We give the model $n = 256$ test problems to solve. In (a), for `No Reverse Path`, we train the model without computing an explicit reverse path, only using the last step of the noising process as targets. For `No Current Image`, we train a model that does not get to see the compiled output image of the program it is editing. For `No Noising`, instead of using our noising process, we generate two random expressions and use the path between them as targets. In (b) we examine the effect of training mixture between forward diffusion ($\rho = 0.0$) and pure random initialization ($\rho = 1.0$) further. Error bars show standard deviation across 5 random seeds.

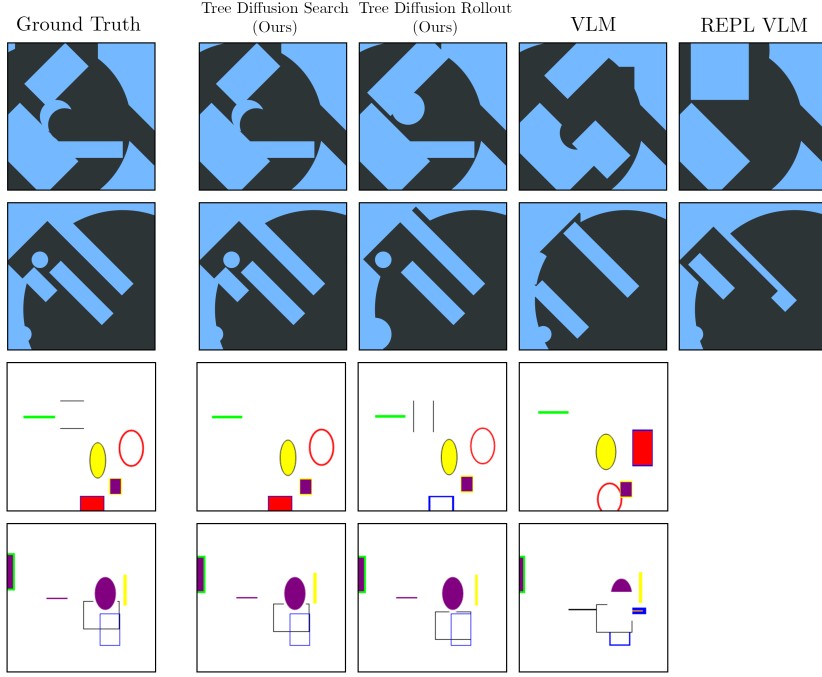

Figure 6: Qualitative examples of our method and baselines on two inverse graphics languages, `CSG2D` (top two rows) and `TinySVG` (bottom two rows). The leftmost column shows the ground-truth rendered programs from our test set. The next columns show rendered programs from various methods. Our methods are able to finely adjust and match the ground-truth programs more closely.

mal intermediate states. Figure 5(a) demonstrates that utilizing the reverse mutation path improves performance, particularly in finding solutions with fewer steps. However, both methods eventually reach similar performance levels, suggesting that a noisy path, while less efficient, can still lead to a solution.

Finally, we explored whether the incremental noise process is crucial, given our tree edit path algorithm. Couldn't we directly sample two random expressions, calculate the path, and train the network to imitate it? We varied the training data composition between pure forward diffusion ($\rho = 0.0$) and pure random initialization ($\rho = 1.0$) as shown in Figure 5(b). We found that a small proportion ($\rho = 0.2$) of pure random initializations combined with forward diffusion yielded the best results. This suggests that forward diffusion provides a richer training distribution around target points, while random initialization teaches the model to navigate the program space more broadly. The emphasis on fine-grained edits from forward diffusion proves beneficial for achieving exact pixel matches in our evaluations.

## 5 CONCLUSION

In this work, we proposed a neural diffusion model that operates on syntax trees for program synthesis. We implemented our approach for inverse graphics tasks, where our task is to find programs that would render a given image. Unlike previous work, our model can construct programs, view their output, and in turn edit these programs, allowing it to fix its mistakes in a feedback loop. We quantitatively showed how our approach outperforms our baselines at these inverse graphics tasks. We further studied the effects of key design decisions via ablation experiments.

**Limitations** There are several significant limitations to this work. First, we operate on expressions with no variable binding, loops, strings, continuous parameters, etc. While we think our approach can be extended to support these, it needs more work and careful design. Current large-language models can write complicated programs in many domains, while we focus on a very narrow task.

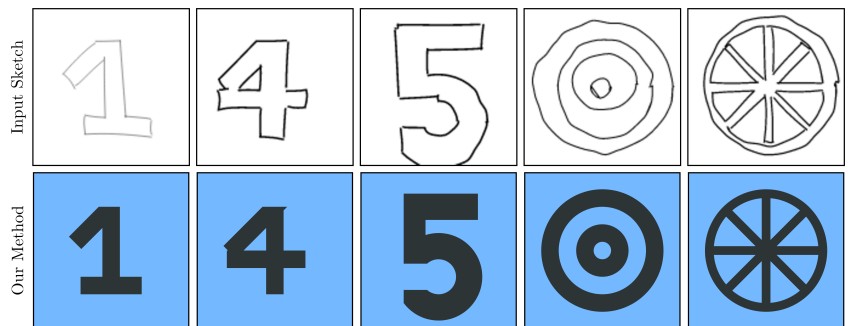

Figure 7: Examples of programs recovered for input sketches in the `CSG2D-Sketch` language. The input sketches are from our observation model that simulates hand-drawn sketches (top-row). The output programs rendered (bottom row) are able to match the input sketches by adding and subtracting basic shapes. Video results for these sketches can be found at `https://tree-diffusion.github.io`.

Additionally, the task of inverse graphics might just be particularly suited for inverse graphics where small mutations make informative changes in the image output.

**Future Work**    In the future, we hope to be able to leverage large-scale internet data on programs to train our system, making small mutations to their syntax tree and learning to invert them. We would also like to study this approach in domains other than inverse graphics. Additionally, we would like to extend this approach to work with both the discrete syntax structure and continuous floating-point constants.

**Impact**    Given the narrow scope of the implementation, we don't think there is a direct societal impact, other than to inform future research direction in machine-assisted programming. We hope future directions of this work, specifically in inverse graphics, help artists, engineering CAD modelers, and programmers with a tool to convert ideas to precise programs for downstream use quickly.

ACKNOWLEDGMENTS

We would like to thank Kathy Jang, David Wu, Cam Allen, Sam Toyer, Eli Bronstein, Koushik Sen, and Pieter Abbeel for discussions, feedback, and technical support. Shreyas was supported in part by the AI2050 program at Schmidt Futures (Grant G-22-63471). Erik was supported by fellowships from the Future of Life Institute and Open Philanthropy.

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

# APPENDIX

## A MUTATION ALGORITHM

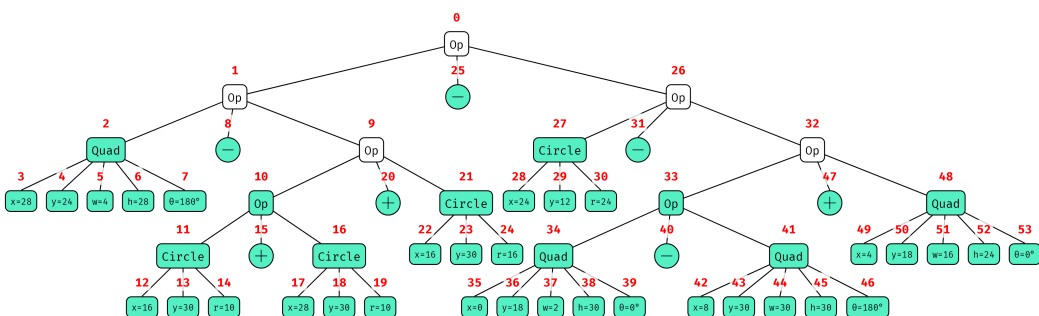

Figure 8: An example expression from `CSG2D` represented as a tree to help illustrate the mutation algorithm. The green nodes are candidate nodes with primitives count $\sigma(z) \leq 2$. Our mutation algorithm only mutates these nodes.

Here we provide additional details on how we sample small mutations for tree diffusion. We will first repeat the algorithm mentioned in Section 3 in more detail.

Our goal is to take some syntax tree and apply a small random mutation. The only type of mutation we consider is a replacement mutation. We first collect a set of *candidate* nodes that we are allowed to replace. If we select a node too high up in the tree, we end up replacing a very large part of the tree. To make sure we only change a small part of the tree we only select nodes with $\leq \sigma_{small}$ primitives. In Figure 8, if we set $\sigma_{small} = 2$, we get all the **green** nodes. We sample a node, $m$,

uniformly from this **green** set. We know the production rule for $m$ from the CFG. For instance, if we selected node $15$, the only replacements allowed are $+$ or $-$. If we selected node $46$, we can only replace it with an angle. If we selected node $11$, we can replace it with any subexpression. When we sample a replacement, we ensure that the replacement is $\leq \sigma_{\text{small}}$, and that it is different than $m$. Here we show $4$ random mutation steps on a small expression,

```
(+ (+ (+ (Circle A D 4) (Quad F E 4 6 K)) (Quad 3 E C 2 M)) (Circle C 2 1))
        ^^^^^^^^^^^^^^^^^^^^^^^^^^^^^^^^^^^^^^^^ --> (Circle 0 8 A)
(+ (+ (Circle 0 8 A) (Quad 3 E C 2 M)) (Circle C 2 1))
      ^^^^^^^^^^^^^^^^^^^^^^^^^^^^^^^^^^^^^^^ --> (Quad 1 0 A 3 H)
(+ (Quad 1 0 A 3 H) (Circle C 2 1))
                           ^ --> 4
(+ (Quad 1 0 A 3 H) (Circle 4 2 1))
                           ^ --> 8
(+ (Quad 1 0 A 3 H) (Circle 8 2 1))
```

During our experiments we realized that this style of random mutations biases expression to get longer on average, since there are many more leaves than parents of leaves. This made the network better at going from very long expressions to target expressions, but not very good at editing shorter expressions into longer ones. This also made our model's context window run out frequently when expressions got too long. To make the mutation length effects more uniform, we add a slight modification to the algorithm mentioned above and in Section 3.

For each of the candidate nodes, we find the set of production rules for the candidates. We then select a random production rule, $r$, and then select a node from the candidates with the production rule $r$, as follows,

$$C = \{n \in \text{SyntaxTree}(z) \mid \sigma(n) \leq \sigma_{\text{small}}\}$$
$$R = \{\text{ProductionRule}(n) \mid n \in C\}$$
$$r \sim \text{Uniform}[R]$$
$$M = \{n \in C \mid \text{ProductionRule}(n) = r\}$$
$$m \sim \text{Uniform}[M]$$

For CSG2D, this approach empirically biased our method to make expressions *shorter* $30.8\%$, equal $49.2\%$, and longer $20.0\%$ of the times ($n = 10,000$).

## B CONTEXT-FREE GRAMMARS

Here we provide the exact context-free grammars of the languages used in this work.

### B.1 CSG2D

```
s: binop | circle | quad
binop: (op s s)
op: + | -

number: [0 to 15]
angle: [0 to 315]

// (Circle radius x y)
circle: (Circle r=number x=number y=number)

// (Quad x y w h angle)
quad: (Quad x=number y=number
            w=number h=number
            theta=angle)
```

### B.2 TINYSVG

```
s: arrange | rect | ellipse | move
```

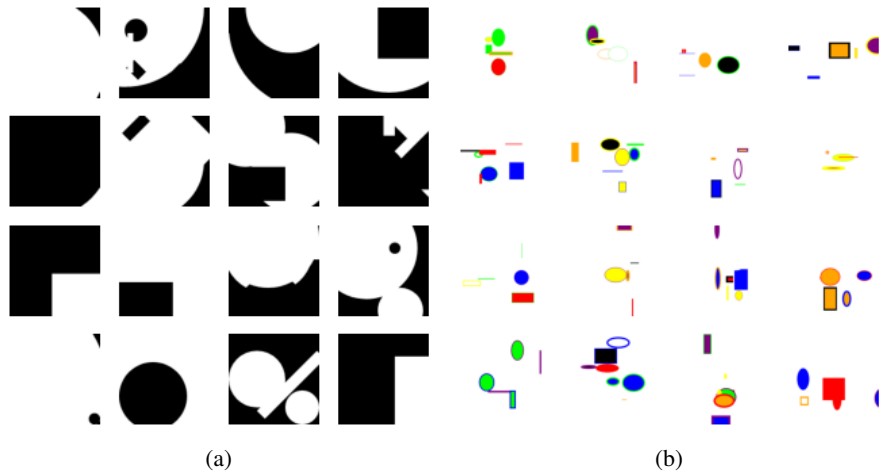

(a)                                     (b)

Figure 9: Examples of images drawn with the (a) `CSG2D` and (b) `TinySVG` languages.

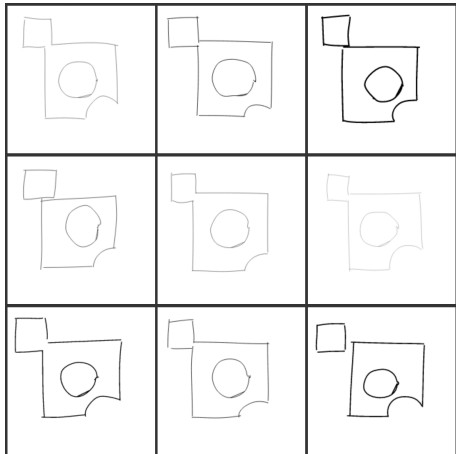

Figure 10: Examples of the same scene being called multiple times by our sketch observation model.

```
direction: v | h
color: red | green | blue | yellow | purple | orange | black | white | none
number: [0 - 9]
sign: + | -

rect: (Rectangle w=number h=number fill=color stroke=color border=number)

ellipse: (Ellipse w=number h=number fill=color stroke=color border=number)

// Arrange direction left right gap
arrange: (Arrange direction s s gap=number)

move: (Move s dx=sign number dy=sign number)
```

## C  SKETCH SIMULATION

As mentioned in the main text, we implement the `CSG2D-Sketch` environment, which is the same as `CSG2D` with a hand-drawn sketch observation model. We do this to primarily show how this sort of a generative model can possibly be applied to a real-world task, and that observations do not

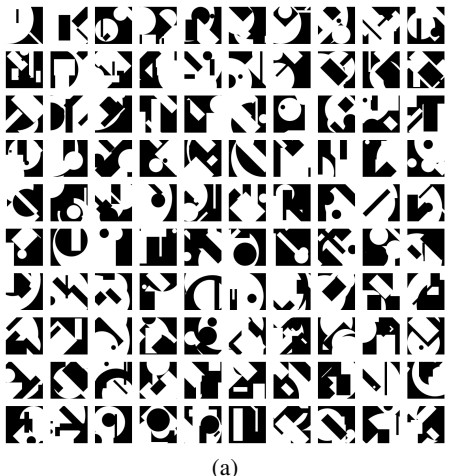 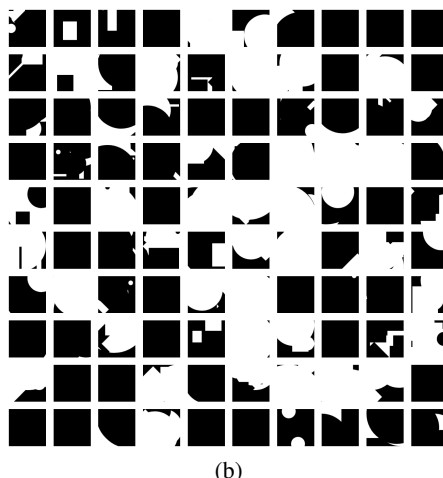

(a) (b)

Figure 11: Examples of thresholding scene images using the LZ4 compression algorithm. The left represents our test set, the right represents our training distribution.

need to be deterministic. Our sketch algorithm can be found in our codebase, and is based off the approach described in Wood et al. (2012).

Our compiler uses Iceberg (IceBerg Contributors, 2023) and Google's 2D Skia library to perform boolean operations on primitive paths. The resulting path consists of line and cubic bézier commands. We post-process these commands to generate sketches. For each command, we first add Gaussian noise to all points stated in those commands. For each line, we randomly pick a point near the 50% and 75% of the line, add Gaussian noise, and fit a Catmull & Rom (1974) spline. For all curves, we sample random points at uniform intervals and fit Catmull-Rom splines. We have a special condition for circles, where we ensure that the start and end points are randomized to create the effect of the pen lifting off. Additionally we randomize the stroke thickness.

Figure 10 shows the same program rendered multiple times using our randomized sketch simulator.

## D  COMPLEXITY FILTERING

As mentioned in Section 4, while testing our method alongside baseline methods, we reached ceiling performance for all our methods. Ellis et al. (2019) got around this by creating a "hard" test case by sampling more objects. For us, when we increased the number of objects to increase complexity, we saw that it increased the probability that a large object would be sampled and subtract from the whole scene, resulting in simpler scenes. This is shown by Figure 11(b), which is our training distribution. Even though we sample a large number of objects, the scenes don't look visually interesting. When we studied the implementation details of Ellis et al. (2019), we noticed that during random generation of expressions, they ensured that each shape did not change more that 60% or less than 10% of the pixels in the scene. Instead of modifying our tree sampling method, we instead chose to rejection sample based on the compressibility of the final rendered image.

## E  TREE PATH ALGORITHM

Algorithm 1 shows the high-level pseudocode for how we find the first step of mutations to transform tree $A$ into tree $B$. We linearly walk down both trees until we find a node that is different. If the target node is *small*, i.e., its $\sigma(z) \leq \sigma_{\text{small}}$, then we can simply mutate the source to the target. If the target node is larger, we sample a random small expression with the correct production rule, and compute the path from this small expression to the target. This gives us the *first step* to convert the source node to the target node. Repeatedly using Algorithm 1 gives us the full path to convert one

---

**Algorithm 1** TreeDiff: Find the first set of mutations to turn one tree to another.

---

**Require:** `treeA`: source tree, `treeB`: target tree, `max_primitives`: maximum primitives
**Ensure:** List of mutations to transform `treeA` into `treeB`
 1: **if** `NodeEq(treeA, treeB)` **then**
 2:   `mutations ← []`
 3:   **for** each `(childA, childB)` in `zip(treeA.children, treeB.children)` **do**
 4:     `mutations ← mutations + TreeDiff(childA, childB, max_primitives)`
 5:   **end for**
 6:   **return** `mutations`
 7: **else**
 8:   **if** `treeA.primitive_count ≤ max_primitives` **and** `treeB.primitive_count ≤ max_primitives` **then**
 9:     **return** `[Mutation(treeA.start_pos, treeA.end_pos, treeB.expression)]`
10:   **else**
11:     `new_expression ← GenerateNewExpression(treeA.production_rule, max_primitives)`
12:     `tightening_diffs ← TreeDiff(new_expression, treeB, max_primitives)`
13:     `new_expression ← ApplyAllMutations(new_expression, tightening_diffs)`
14:     **return** `[Mutation(treeA.start_pos, treeA.end_pos, new_expression)]`
15:   **end if**
16: **end if**

---

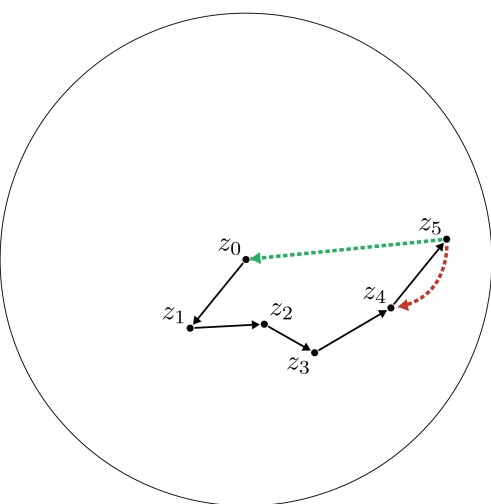

Figure 12: A conceptual illustration of why we need tree path-finding. The red path represents the naive target for the neural network. The green path represents the path-finding algorithm's target.

expression to another. We note that this path is not necessarily the optimal path, but a valid path that is less noisy than the path we would get by simply chasing the last random mutation.

Figure 12 conceptually shows why computing this tree path might be necessary. The circle represents the space of programs. Consider a starting program $z_0$. Each of the black arrows represents a random mutation that *kicks* the program to a slightly different program, so $z_0 \rightarrow z_1$, then $z_2 \rightarrow z_3 \ldots$. If we provide the neural network the supervised target to go from $z_5$ to $z_4$, we are teaching the network to take an inefficient path to $z_0$. The green path is the direct path from $z_5 \rightarrow z_0$.

| Method | % Test Tasks Solved |
|---|---|
| Our Approach | 32.18% |
| LLaVA (Base) | 0.00% |
| LLaVA (Base+SVG Prompt) | 0.00% |
| LLaVA (Fine-Tuned) | 1.17% |

Table 1: Performance of LLaVA on `CSG2D` test tasks. Note that both approaches were given a budget of 100 samples from the model to complete these tasks, which is lower than the evaluation results reported earlier. This is because the LLaVA model has considerably more parameters and is much slower to run.

## F    COMPARISON WITH LARGE PRETRAINED MODELS

We perform some additional experiments to compare the performance of modern large vision-language models that are pretrained on internet scale data with our approach. Specifically we evaluate the performance of LLaVA (Liu et al., 2024) on the `CSG2D` task. The base model performs $0\%$ on our test set (100 rejection samples), and this makes sense since it is not a fair comparison, it has not seen our DSL. Interestingly, LLaVA gets $0\%$ even when asked to write SVG programs, something that it has been trained on. This is because passing requires the output image to be very close to the required specification.

To make it more fair for LLaVA, we fine-tuned it using the author's original `finetune_task` script alongside the default suggested hyperparameters. To test LLaVA, we provide it with the test image and rejection sample its output 100 times. We used `LLaVA-1.5-7B`, and trained on a single A100 graphics card. Our model was also given 100 node-expansions in the tree search.

The LLaVA model has 7B parameters, and our model uses approximately 700x fewer parameters. Because of this, node expansion count is no longer an equivalent metric for computation. If we let our method use the same wall-clock time on the same GPU, our method reaches the ceiling performance of $84.68\%$.

Figure 11 shows just how hard our task is, the left are examples from the test set, the right are examples from the training distribution. Having a strict requirement of matching the specification makes it a very demanding task.

## G    IMPLEMENTATION DETAILS

We implement our architecture in PyTorch (Ansel et al., 2024). For our image encoder we use the NF-ResNet26 (Brock et al., 2021) implementation from the open-sourced library by Wightman (2019). Images are of size $128 \times 128 \times 1$ for `CSG2D` and $128 \times 128 \times 3$ for `TinySVG`. We pass the current and target images as a stack of image planes into the image encoder. Additionally, we provide the absolute difference between current and target image as additional planes.

Our decoder-only transformer (Vaswani et al., 2017; Radford et al., 2019) uses 8 layers, 16 heads, with an embedding size of 256. We use batch size 32 and optimize with Adam (Kingma & Ba, 2014) with a learning rate of $3 \times 10^{-4}$. All models, including baselines were trained for 1 million steps. The image embeddings are of the same size as the transformer embeddings. We use 4 prefix tokens for the image embeddings. We used a maximum context size of 512 tokens. For both environments, we sampled expressions with at most 8 primitives. Our method and all baseline methods used this architecture. We did not do any hyperparameter sweeps or tuning.

For the autoregressive (VLM) baseline, we trained the model to output ground-truth programs from target images, and provided a blank current image. For tree diffusion methods, we initialized the search and rollouts using the output of the autoregressive model, which counted as a single node expansion. For our re-implementation of Ellis et al. (2019), we flattened the `CSG2D` tree into shapes being added from left to right. We then randomly sampled a position in this shape array, compiled the output up until the sampled position, and trained the model to output the next shape using constrained grammar decoding.

This is a departure from the pointer network architecture in their work. We think that the lack of prior shaping, departure from a graphics specific pointer network, and not using reinforcement learning to fine-tune leads to a performance difference between their results and our re-implementation. We note that our method does not require any of these additional features, and thus the comparison is fairer. For tree diffusion search, we used a beam size of $64$, with a maximum node expansion budget of $5000$ nodes.

The value network is a small 2-layer MLP with 128 units in each layer that takes in the output of the vision encoder, and outputs a single scalar predicting the edit distance in program space between two images. We train this for $100,000$ steps.

