# OpenReview forum: "Diffusion On Syntax Trees For Program Synthesis"
_ICLR.cc/2025/Conference — ICLR 2025 Spotlight_

### Official Review · Reviewer_ZXAr · 2024-10-31

**Soundness:** 3
**Presentation:** 4
**Contribution:** 3
**Rating:** 8
**Confidence:** 5

**Summary:**

This paper addresses the edge of visual symbolic reasoning and code generation. It addresses the important task of generating code (symbolic sequence) to depict images with visual feedbacks. It applies diffusion model-like approaches to permute the program syntax tree and guarantee the correctness of the generated code. Through iterations, the model is able to recover the image with high fidelity using discrete preset symbols.

**Strengths:**

- This paper proposed a novel solution to the reverse CG field, to synthesis programs for visual symbolic reasoning. The proposed method address the hard task through the unique lens of syntax tree, and achieves notably better results.

- The idea of permuting on syntax tree allows for more efficient model, with better performance.

- The efforts to make demo video makes the paper easier to understoand and spread.

**Weaknesses:**

This is a good paper, with minor weakness points below.

- It is better to mention the size of the decoder model in the architecture section rather than in the appendix, so that readers with LLM background can quickly understand the edge of the model on this task.

- It is better to discuss the number of steps in the diffusion procedure, and the model's potential ability limit in terms of output sequence length or number of symbols.

- Two highly related work should be cited and discussed:

"Outline, Then Details: Syntactically Guided Coarse-To-Fine Code Generation" in ICML 2023, which explore the possibility of syntax tree to generate code, and via coarse-to-fine multi-round generation approach.

"Symbolic Visual Reinforcement Learning: A Scalable Framework with Object-Level Abstraction and Differentiable Expression Search" in TPAMI, which also learns visual symbolic programs, not to depict the image but to interact with the environments. Rainbow environment is also leveraged in their experiments.

**Questions:**

See discussions above.

---

> ### Author Response · Authors · 2024-11-14
>
> Thank you for your positive review, we are very encouraged by your positive feedback and comments. We also thank you for providing insightful discussion and thoughts about our work.
>
> > It is better to mention the size of the decoder model in the architecture section rather than in the appendix, so that readers with LLM background can quickly understand the edge of the model on this task.
>
> This is a really good point, and we will state this in the main text. For reference, the model we are using is much smaller than typical LLMs, around ~10 million parameters.
>
> > It is better to discuss the number of steps in the diffusion procedure, and the model's potential ability limit in terms of output sequence length or number of symbols.
>
> We wanted to ask for clarification to this point, is this asking us to measure performance conditioned on the number of symbols in the ground-truth test tasks? So, fewer symbols should be easier and more symbols should be harder? We discovered that increasing the number of symbols ended up making the test tasks easier than harder, since the probability we would subtract a big shape to make the image fully black was increased, so we went for the procedure in Appendix D.
>
> > Two highly related work should be cited and discussed …
>
> Both of these works are very relevant, and we will cite these works. Thank you for bringing these to our attention.
>
> ---
>
> Thank you for your feedback, your positive review, and insightful questions.

---

### Official Review · Reviewer_GpLx · 2024-11-01

**Soundness:** 3
**Presentation:** 3
**Contribution:** 2
**Rating:** 6
**Confidence:** 3

**Summary:**

The paper presents an algorithm to train a diffusion model on the abstract syntax tree (AST) of programs written in a simple procedural language. That language produces 2D images by combining geometric shape primitives, and the resulting image is used to guide the denoising process.

Using an additional network estimating the distance from a rendered image to the target image, beam search is applied to generate a sequence of AST edits (node replacements) that produce a predicted program, approximating the target image. That approach transfers to generating geometric images from noisy sketches.

**Strengths:**

Originality
-------------
The paper takes inspiration from existing ideas and benchmarks, but they are clearly cited, and the novel aspects are well described. For instance, a backward edit path that's better than reversing the corruption path, removing the need of a partial renderer, and relying on beam search rather than full-fledged reinforcement learning.

Quality
----------

Experiments demonstrate the advantages of the proposed approach, and properly ablate the different aspects and contributions.

Clarity
---------
The paper is overall clear and straightforward to follow. With the additional details of the appendix, the approach should be re-implementable by a different team.

Significance
-----------------
Using ML models to directly manipulate and modify programs, rather than either generate a whole program autoregressively, or emit edition instructions (which could be invalid or result in an invalid program) could make iterative program generation better or easier.
The fact that no reinforcement learning is required, but observation of the output of intermediate programs can simply be combined with beam search is also an interesting result.

**Weaknesses:**

Originality
-------------
No major weakness here, the work is in the continuation of previous cited work.

Quality
----------
Comparison with baselines might have been more extensive, specifically the RL-based algorithms from previous work, which could have better shown how "brittle" they were.

Clarity
---------
A few things were not clearly defined in the experiments and ablation sections (see "questions" below).

Significance
----------------
Overall, the CSG2D and TinySVG languages are a small-scale benchmark, but it's unclear whether the proposed approach would scale to large, structured programs in general purpose languages.
For instance, it might not be possible to find a sequence of valid programs created by short mutations between two relatively close programs. For instance, going from recursion to a loop, from an implicit lamda to a declared function, or from a for loop to a list comprehension. Even splitting a function into smaller pieces may require either large edits, or intermediate unparseable states.

After discussion
----------------------
The authors provided clarifications, other reviewers raised a few additional concerns, overall I'm maintaining my score.

**Questions:**

1. In the ablations, there is one about training only on the last mutation step. It's illustrated by Fig. 12 by only showing the transition from z_5 to z_4. Do I understand correctly that the other reverse transitions (e.g., z_4 -> z_3, ..., z_1 -> z_0) are not used for training?
2. If so, why not use that (training on all denoising steps) as a baseline?
3. Can you define the "Rollout" method, and the differences with the "Search" one?
4. What's the relationship between "Number of nodes expanded" and the "number of compilations needed"?

---

> ### Author Response · Authors · 2024-11-14
>
> Thank you for your encouraging review. We are glad that you see the removal of the partial compiler and having to use no reinforcement learning while still using the execution output as strengths of our approach. We will answer your insightful questions here.,
>
> > In the ablations, there is one about training only on the last mutation step. It's illustrated by Fig. 12 by only showing the transition from z_5 to z_4. Do I understand correctly that the other reverse transitions (e.g., z_4 -> z_3, ..., z_1 -> z_0) are not used for training?
>
> This is correct, the other transitions are not used for training. We considered using all the transitions at once as a training signal, but we were not actually sure how to do that, since the network must receive the execution output of the “current” program, which program would be the “current”?
>
> > Can you define the "Rollout" method, and the differences with the "Search" one?
>
> This is our mistake for not stating, we will edit to add this description. Tree Diffusion Rollout is using the “policy” network’s actions (i.e. edit actions) being sampled from the model repeatedly, progressively editing an initial program, $z_0 \to z_1 \to \ldots$. Tree Diffusion Search is using this same policy, but instead sampling multiple possible “edits” and ranking them using the value network, as in beam search, which we described in Section 3.3.
>
> > What's the relationship between "Number of nodes expanded" and the "number of compilations needed"?
>
> This is indeed a typo in Line 372, we will update this to be consistent and say “Number of nodes expanded” instead of “number of compilations needed”. Each node is a full program that is either edited through pure policy (rollouts), via search (policy+value), or by repeatedly re-sampling the node from the VLM baseline. We will fix this.
>
> ---
>
> We thank you for your review, your feedback, and thoughtful discussion about our work.

---

> > ### Comment · Reviewer_GpLx · 2024-11-26
> >
> > > the other transitions are not used for training. We considered using all the transitions at once as a training signal, but we were not actually sure how to do that, since the network must receive the execution output of the “current” program, which program would be the “current”?
> >
> > I'm not sure what the problem would be, since you should be able to render the image at each of the intermediate stages (z_1, ..., z_4), as represented on Figure 2. Then there would be as many examples as mutation stages: predict $(z_{t-1} | z_t, x_t, x_{t-1})$.
> >
> > I think it would be a more reasonable baseline, otherwise the model is never actually trained to reverse the first mutation.
> >
> > In fact it raises another question: In the ablation, did you train the model to predict $(z_4 | z_5, x_5, x_4)$? Or did you use $x_0$? If you did, I'm not sure that's a strong enough baseline to show that the path-finding algorithm is necessary.

---

> > > ### Author Response · Authors · 2024-11-26
> > >
> > > Apologies, we misunderstood your question. We missed the part "In the ablations", and thought you were asking about the proposed model, not the ablated one.
> > >
> > > > otherwise the model is never actually trained to reverse the first mutation.
> > >
> > > The number of mutations $t$ is randomly sampled from a uniform distribution, $t \sim U[1, T]$, so sometimes we only do a single mutation, sometimes more. We train the ablated model to predict $(z_{t-1} | z_{t}, x_{t}, x_{0})$. But since $t$ (the maximum number of mutations) can sometimes be just a single mutation, it is in fact trained to reverse the first mutation.
> > >
> > > > as many examples as mutation stages: predict $(z_{t-1} | z_{t}, x_{t}, x_{t-1})$
> > >
> > > Wouldn't this model only be capable of reverse noise for a single step then? If provided with a test image, it would only be able to "heal" programs that are one edit step away, but any more would be out of distribution.
> > >
> > > Please let us know if we misunderstood again, we really appreciate your time and discussion.

---

### Official Review · Reviewer_4cr6 · 2024-11-04

**Soundness:** 3
**Presentation:** 2
**Contribution:** 3
**Rating:** 6
**Confidence:** 3

**Summary:**

This paper proposed a program synthesis framework using mutations on syntax trees via a neural diffusion model for inverse graphic tasks. These tasks aim to convert images that can contain free hand-sketches of shapes or a set of computer-generated colored shapes into images depicting a computer-generated rendering matching the input. The authors defend the claim that the approach presents the ability to edit trees generated using a base model as opposed to incrementally autoregressive approaches that fail to narrow down the search space. The authors apply their method to inverse graphics tasks and present results in two settings (CSG2D and TinySVG) and show improved performance in the number of problems solved compared to baselines (REPL VLM and VLM) along with an ablation study to investigate the individual contributions of constituent components.

**Strengths:**

1. The main strength of this paper is the design of a neurosymbolic framework to evaluate the automated (i.e. diffusion-based) conversion of images into context-free grammar. This formal evaluation ensures that the desired specifications are met through iterative observation of the execution results and verification.

2. The authors extend the approach to accept hand-drawn sketches and illustrate examples in the appendix confirming the applicability of the approach in several real-world settings.

3. The supplementary videos illustrate the overall problem that the authors are attempting to solve and showcases the "edits" made by the framework.

**Weaknesses:**

There are three main weaknesses I would like to bring up. The authors are encouraged to rebut and provide legitimate explanations, if any, against these and the review decision may be adjusted accordingly.

1. A claim made by the author states that the proposed method focuses on editing the program synthesized from the image, unlike prior works that autoregressively generate programs that are incrementally better. In doing so, the authors propose adding random noise to modify a base syntax tree generated from CSG2D. Despite the illustrative example shown in Figure 2, enabling the approach to modify node types rather than shape, the base syntax tree structure is governed by the initial generated program. It remains unclear (at least it has not been proven) that diffusion + base tree always yields the optimal syntax tree (a statement regarding suboptimal steps in section 4.3 is thus not justified). An analysis and example to demonstrate this is lacking and should be included.

2. The overall architecture presented in Figure 3 is difficult to understand at first glance. In addition, the descriptions provided in section 3.4 (the model architecture) do not present enough detail to understand Figure 3.  Specifically, it is not clear how replacing the "(" denoted by the edit position token and replacing it with the grammar constrained autoregressive decoding yield valid syntax (i.e. are there low-level implementations in play that ensure that entire blocks from “(“ to “)” are parsed out during replacing? How are varying input lengths handled? ). Replacing "(" with "(Quad 8..." seems to break the pairing of parenthesis. In addition, it is not clear what the purpose of "EOS" is in this context.

3. The fraction of problems solved by the method trained with "no reverse path" is nearly the same as that of the control after about 60 expanded nodes. The control reaches the same performance at about 50 nodes. Is this a "significant" efficiency gain when the maximum node expansion budget was two orders of magnitude higher (i.e. 5000)? There are no computational or time-related metrics presented which help put this into context.

4. Overall the presentation of section 3, especially 3.4, requires careful rework.

**Questions:**

Here are some of the main points I would like to make:

1. The edit distance introduced by Pawlik & Augsten is narrowed to allow only small changes. If there are big changes, the change that reduces the distance between the trees the most is chosen. While this can be used as a training signal, it is assumed and was mentioned in section 3.3.2 that access to the ground-truth mutations is available. However, access to this ground truth may not be readily available in most cases. How is this ground-truth data obtained? If not automated, the authors must comment on the limited scalability of the approach.

2. In section 4.2 under evaluation, it is not clear what the criteria for considering a match between the synthesized and true plan is. Re: “In TinySVG, we accepted an image if 99% of the pixels were within 0.005 ≈ $\frac{1}{256}$.” Is this “within 0.005” a tolerance on the 8-bit pixel color intensity? If so, please state explicitly and explain how this metric is applied to the RGB images in TinySVG.

3. The difference between tree diffusion search and tree diffusion rollouts is not explicitly stated or defined in section 4.

4. There were references to computational demand and efficiency, yet no time-related metrics were reported to demonstrate gains in this regard, despite claims about improving performance efficiency. It is unclear as to just how much improvement the proposed approach affords.

---

> ### Author Response · Authors · 2024-11-14
>
> Thank you for your insightful review, we really appreciate the thorough reading alongside your detailed and engaged questions. We are also encouraged that you found our neurosymbolic design to be a strength of our work. We will attempt to address your mentioned weaknesses and concerns here.
>
> >  the illustrative example shown in Figure 2, enabling the approach to modify node types rather than shape, the base syntax tree structure is governed by the initial generated program.
>
> We will try to illustrate how our mutation scheme allows for not only changing node types, but also the shape and structure of the whole syntax tree here. Consider the tree in Figure 8, in Appendix A. Under our approach, all nodes in green are potential mutation targets. Node 44 is the width of the rectangle and has type `Number`. And it can be replaced with any other `Number`. This allows changing node types. For changing tree shape and structure, consider Node 33. Node 33 contains 2 primitives, the left Quad and the right Quad. A mutation target can replace this _entire_ node with a single node that contains only 1 primitive. Let’s say we replace Node 33 with a Circle node. We have effectively changed the shape of the tree. With this Circle node, Node 32 is now also a potential mutation target, and can be replaced with a smaller subtree and so on. This way, we can actually delete the whole tree with our mutations. The reverse is also true, a node with just one primitive can be replaced with a node with two, which allows us to “grow” any tree using a series of small mutations.
>
> We encourage the reviewer to look at our interactive demo for this (https://td-anon.github.io/). If you scroll down on the page and click the “Add Noise” button, it will run our code in the browser to show how our mutations can edit both the type _and_ shape of the tree.
>
> If this makes it clearer, we will include an illustrative description such as above to make this important point clearer.
>
> > Specifically, it is not clear how replacing the "(" denoted by the edit position token and replacing it with the grammar constrained autoregressive decoding yield valid syntax …
>
> The figure is indeed confusing in this way. It looks like it’s replacing the opening “(“ with the whole expression, but your hunch is correct, every parenthesis actually uniquely refers to a node in the syntax tree (or alternatively every opening “(“ is implicitly matched with a closed “)” and the entire sub-expression is replaced). These details are actually very nicely handled by the fact that we are using a context-free grammar and the excellent Lark library for parsing, which automatically maps the node in the syntax tree to tokens in the S-expression representation. Our implementation is flexible such that only the grammar needs to be changed and it would work with other formats.
>
> > How are varying input lengths handled? …  In addition, it is not clear what the purpose of "EOS" is in this context.
>
> The Transformer architecture we’re using (akin to GPT-2) can handle varying input lengths (in the same way as large language models). The sequences are modeled autoregressively, and hence the `<EOS>` (i.e. “end of sequence”) token is used for the model to indicate that it has stopped generating. We could just stop decoding when all syntax constraints are met, however, using the `<EOS>` token allows us to re-use standard Python libraries that work with these Transformer models.
>
> > The fraction of problems solved by the method trained with "no reverse path" is nearly the same as that of the control after about 60 expanded nodes.
>
> Again, great catch! We will remove the word “significantly” here. It does improve the results, but as you mentioned, not by much. We actually think this is a strength, since computing the optimal reverse path is not always possible, and the experiment shows that it is not strictly needed. We will restructure the description of this to be more along the lines of “it helps a little but not by much”.
>
> > Overall the presentation of section 3, especially 3.4, requires careful rework.
>
> We are grateful for your feedback on Section 3.4 (based on your previous weaknesses and questions), and will make the issues you bring up clearer.
>
> > However, access to this ground truth may not be readily available in most cases.
>
> The ground-truth mutations are actually just the reverse of the noise mutations. Every mutation can be inverted, so if we add noise mutations, the ground-truth are the reverse of these mutations. We add a variable number of mutations, so the ground-truth distance is the number of mutations we added.
>
> (continued ...)

---

> ### Author Response · Authors · 2024-11-14
>
> (... continued)
>
> > If so, please state explicitly and explain how this metric is applied to the RGB images in TinySVG.
>
> We accept an image if at least 99% of the pixels have every channel (RGB) within $1/256$ of the goal image. For each channel we have 256 possible values, and for a value to be considered the same they have to be within 1 step of each other. So an RGB tuple of $(76, 14, 243)$ is accepted with $(76, 13, 244)$ but not with $(88, 9, 10)$. We want at least 99% of the pixels to match in this way. Since these images are normalized between $0$ and $1$, this threshold is $0.005$.
>
> > The difference between tree diffusion search and tree diffusion rollouts is not explicitly stated or defined in section 4.
>
> Again, this is our mistake for not stating, we will edit to add this description. Tree Diffusion Rollout is using the “policy” network’s actions (i.e. edit actions) being sampled from the model repeatedly, progressively editing an initial program, $z_0 \to z_1 \to \ldots$. Tree Diffusion Search is using this same policy, but instead sampling multiple possible “edits” and ranking them using the value network, as in beam search, which we described in Section 3.3.
>
> > There were references to computational demand and efficiency, yet no time-related metrics were reported to demonstrate gains in this regard, despite claims about improving performance efficiency.
>
> All methods (including baselines) use the same neural architecture. All methods also use the same renderer (compiler). The cost of expanding a node is the cost of a forward pass through the neural architecture + the cost of rendering the program and checking. Therefore the X-axis in Figure 4 represents the total computation needed. For instance, increasing the model parameter count would scale the time taken by each method identically.
>
> ---
>
> We hope we are able to clarify some of your concerns, and please let us know if we misunderstood any point. We really appreciate the feedback, both positive and negative, and the insightful discussions.

---

> ### Comment · Reviewer_4cr6 · 2024-11-27
>
> 1. Yes, a detailed explanation via an example is a must. However, as I had mentioned earlier, it remains unclear (at least it has not been proven) that diffusion + base tree always yields the optimal syntax tree (a statement regarding suboptimal steps in section 4.3 is thus not justified).
>
> 2. Given the figure is confusing, which the authors agree with as well, this needs to be addressed and made clear.
>
> 3. Yes, declaiming the "significant" would eliminate the exageration. Yet, as I mentioned, there are no computational or time-related metrics presented which help put this into context.
>
> 4. Re: tree search vs rollouts - such clarification is warranted for the reader to understand the differences.
>
> 5. Re: time of computations, use of the same network architecture should be explicity stated for the reader to grasp this.

---

> > ### Author Response · Authors · 2024-11-27
> >
> > Thank you for your response and feedback!
> >
> > > 1. Yes, a detailed explanation via an example is a must.
> >
> > Understood, we include an illustrative example of various tree mutations that can change both shape and type in Appendix A, leveraging Figure 8.
> >
> > > 1. (...) it remains unclear (at least it has not been proven) that diffusion + base tree always yields the optimal syntax tree (a statement regarding suboptimal steps in section 4.3 is thus not justified).
> >
> > What does suboptimal mean in this context? Is this referring to the statement in Section 4.3, "While training on the last mutation step alone provides a valid path, it introduces noise by potentially targeting suboptimal intermediate states."?
> >
> > If it is, then we provide an example of this suboptimality in Section 3.2.2 and an additional illustration with Figure 12. The "optimality" we refer to is in the number of steps to get the target image. As you pointed out earlier, this suboptimality is not critical to the success of our method (Figure 5(a)), but we still think it is interesting from an academic perspective.
> >
> > > 2. Given the figure is confusing, which the authors agree with as well, this needs to be addressed and made clear.
> >
> > We made an update to Figure 3 to highlight the entire subexpression as being mutated, instead of just the "(". Thank you for this feedback!
> >
> > > 4. Re: tree search vs rollouts - such clarification is warranted for the reader to understand the differences.
> >
> > We added these descriptions in the paper based on your feedback. (In Section 4.2, Lines 333-338).
> >
> > > 5. Re: time of computations, use of the same network architecture should be explicity stated for the reader to grasp this.
> >
> > This is also a good point, we have made an update to the caption of Figure 4 and also the description in Section 4.2, Lines 355 onwards to state this explicitly.
> >
> > ---
> >
> > We really value your feedback, and have tried our best to make changes to the paper and add these clarifications. Please let us know if we misunderstood anything. We thank you for your time and discussion with us.

---

### Official Review · Reviewer_6hsj · 2024-11-04

**Soundness:** 4
**Presentation:** 3
**Contribution:** 3
**Rating:** 8
**Confidence:** 3

**Summary:**

The paper introduces an innovative approach to inverse graphics tasks by combining diffusion models with transformers. The authors present the first application of diffusion to program synthesis using explicit syntax tree updates, validating their method on CSG2D and TinySVG environments.

**Strengths:**

- Innovative Approach: The paper presents a novel combination of autoregressive, diffusion, and search methodologies, which, despite being applied to a specific domain, holds potential for broader applications. The reverse mutation path algorithm also provides an efficient way to generate training targets.
- Clarity and Replicability: The manuscript is well-written and easy to follow, providing sufficient detail to enable replication of the experiments.
- Comprehensive Ablation Studies: The authors conduct thorough ablation studies on key hyperparameters and the impact of integrating search, enhancing the understanding of their method's efficacy.

**Weaknesses:**

- Literature Coverage: The authors should consider citing "Outline, Then Details: Syntactically Guided Coarse-To-Fine Code Generation" in the Neural program synthesis section since this work also takes multiple passes of the program and edits the program.
- The value network (vϕ) training and effectiveness aren't thoroughly evaluated. Alternative approaches to edit distance estimation, including direct calculation from syntax trees, are not explored or compared.

**Questions:**

- In the limitations section, you mention that your syntax tree currently supports only a limited set of operators. What are the bottlenecks in expanding support to other operators and generalizing to broader coding problems?
- What is the cost of training the value network that predicts the edit distance?
- Given the recent advances in vision-language models, how does your approach compare against contemporary models like VILA or LLaMA? The current baselines only include older models (4+ years old), and evaluating against recent state-of-the-art would provide a more helpful comparison.

---

> ### Author Response · Authors · 2024-11-15
>
> Thank you for your review, and the strengths you identified and also the insightful feedback. We are glad you found our paper innovative, clear, reproducible and sound.
>
> > The authors should consider citing "Outline, Then Details: Syntactically Guided Coarse-To-Fine Code Generation" in the Neural program synthesis section since this work also takes multiple passes of the program and edits the program.
>
> This is really good feedback, we will cite this paper and it is indeed very relevant.
>
> > The value network (vϕ) training and effectiveness aren't thoroughly evaluated. Alternative approaches to edit distance estimation, including direct calculation from syntax trees, are not explored or compared.
>
> During inference time, we don’t have access to the ground truth syntax tree of the goal image, so we can’t directly compute the edit distance. To that end we need a value network to provide an estimate of progress to be used in tandem with search. Removing the value network from search would reduce it to doing just policy rollouts which we study in Figure 4. The value network is trained via supervised learning by directly calculating the edit distance between syntax trees like you mentioned. We agree that the specifics of the value network are not discussed in detail, to which we will make changes for clarity. We apologize if we misunderstood this point, and please let us know if you meant something else or if there is an ablation or experiment we could do to strengthen our results!
>
> > In the limitations section, you mention that your syntax tree currently supports only a limited set of operators. What are the bottlenecks in expanding support to other operators and generalizing to broader coding problems?
>
> This is a really good question, and something we can expand on in the limitations section. There are a couple of challenges to applying this to general purpose languages, which we actively want to work on for future work:
>
> For graphics programs, the execution output is self contained, an image with a predictable size, which makes it easier to study. For general purpose programs, not only do we need to provide the final execution output, we also need to provide debugging _traces_ for variables within the program to effectively make use of the execution. Ideally we actually want to do these editing passes on computational _graphs_ instead of trees. It is not entirely obvious how to perform random mutations on graphs that make the program executable, and also only change a small part of the program.
>
> > What is the cost of training the value network that predicts the edit distance?
>
> The value network is a small 2-layer MLP with 128 units in each layer that takes in the output of the vision encoder (NF-ResNet 26), and outputs a single scalar predicting the edit distance in program space between two images. We train this for 100,000 steps. For efficiency, we could have trained this alongside the training of the policy. Again, this is a detail we missed in the paper, and we will make changes to clarify this.
>
> (continued...)

---

> > ### Author Response · Authors · 2024-11-15
> >
> > (... continued)
> >
> > > Given the recent advances in vision-language models, how does your approach compare against contemporary models like VILA or LLaMA? The current baselines only include older models (4+ years old), and evaluating against recent state-of-the-art would provide a more helpful comparison.
> >
> > We used Tsimpoukelli et. al (2021), which is a vision-language model for _all_ our baselines. Our method, and the baseline methods share an identical architecture such that the comparison is fair and the x-axis of Figure 4 represents an identical amount of computation between methods. The method for combining vision and language here was suggested in 2021, but a nearly identical architecture is used by a modern VLM like LLaVA (2023), except for the pretraining, and frozen clip encoder.
> >
> > We performed some additional experiments with a larger pre-trained LLaVA model. The base model performs 0% on our test set (100 rejection samples), and this makes sense since it is not a fair comparison, it has not seen our DSL. Interestingly, LLaVA gets 0% even when asked to write SVG programs, something that it has been trained on. This is because passing requires the output image to be very close to the required specification.
> >
> > To make it more fair for LLaVA, we fine-tuned it using their original `finetune_task` script alongside the default suggested hyperparameters. To test LLaVA, we provide it with the test image and rejection sample its output 100 times. We used LLaVA-1.5-7B , and trained on a single A100 graphics card. Our model was also given 100 node-expansions in the tree search.
> >
> > |Method|% Test Tasks Solved|
> > |---|---|
> > |Our Approach|32.18%|
> > |LLaVA (Base)|0%|
> > |LLaVA (Base+SVG Prompt)|0%|
> > |LLaVA (Fine-Tuned)|1.17%|
> >
> > The LLaVA model has 7B parameters, and our model uses approximately 700x fewer parameters. Because of this, node expansion count is no longer an equivalent metric for computation. If we let our method use the same wall-clock time on the same GPU, our method reaches the ceiling performance of $\approx$ 85%.
> >
> > Figure 11 shows just how hard our task is, the left are examples from the test set, the right are examples from the training distribution. Having a strict requirement of matching the specification makes it a very demanding task.
> >
> > ---
> >
> > Again, we thank you for taking the time to read and review our paper, and also provide insightful feedback, comments, and discussion.

---

### Official Review · Reviewer_JiwH · 2024-11-07

**Soundness:** 3
**Presentation:** 4
**Contribution:** 3
**Rating:** 8
**Confidence:** 4

**Summary:**

The authors propose a program synthesis method based on "tree diffusion". They randomly corrupt programs (with some constraints) and learn to invert the corruptions, conditioned on the output of the corrupted program and the target output (an image rendering in their case).

**Strengths:**

The method is simple but non-obvious
The more general problem of program synthesis conditioned on desired outputs is very relevant
The authors use randomly generated programs as a dataset which sidesteps dataset curation in favor of just a specification of the language
The paper is well-written, easy to understand, and has nice and (mostly) clear figures

**Weaknesses:**

The paper is somewhat limited in scope (simple problem setup) in ways that make it not entirely obvious how the method "scales" to more complex relevant tasks like code generation.

Some minor things covered in Questions

**Questions:**

Fig 3 is confusing. v(x) is the value function or the pre-trained image encoder? if its pretrained, why is there a _phi subscript?

Where do the initial problems come from? It seems like they are generated randomly, but how?

Do the fuzzing and edit algorithms generalize easily to non-context-free grammars (e.g. general programming languages)?

How many steps did you train for? I don't think this is covered in the appendix.

---

> ### Author Response · Authors · 2024-11-14
>
> Thank you for your positive review, we are very encouraged that you thought our paper was simple but non-obvious, well-written, and easy to understand.
>
> > Fig 3 is confusing. v(x) is the value function or the pre-trained image encoder? if its pretrained, why is there a _phi subscript?
>
> You are correct about Figure 3, this is a typo and we will make a correction to this. We were trying to use the same notation as in Tsimpoukelli et al. (2021), where they used $v_\phi$ for the image encoder, but just made the mistake of re-using the notation as the value function. Thank you for pointing this out!
>
> > Where do the initial problems come from? It seems like they are generated randomly, but how?
>
> We (very) briefly touch on this in Section 3.1, Line 183. The initial problems come from our `ConstrainedSample` function, which follows Luke (2000) and Zeller et al. (2023), which randomly samples programs from the context-free grammar. You are correct that we don’t specify that this method is used to generate the problems, and we will definitely clarify this.
>
> > Do the fuzzing and edit algorithms generalize easily to non-context-free grammars (e.g. general programming languages)?
>
> This is a really good question, and we believe it is an open research question for future work. Indeed, there have been fuzzers written for general programming languages (for instance, AFL can fuzz JavaScript code for systems like V8, etc.), but it is not obvious yet how to use that appropriately for an execution guided system like ours. We think this is an exciting future direction!
>
> > How many steps did you train for? I don't think this is covered in the appendix.
>
> We trained all methods, including the baselines for 1 million steps. You are correct that it wasn't covered in the Appendix, and is also something we will add.
>
> ---
>
> Again, we thank you for your encouraging review, insightful questions, and we will take the feedback into account and make the appropriate clarity changes.

---

> > ### Comment · Reviewer_JiwH · 2024-11-26
> >
> > Thanks for the replies!
> >
> > >This is a really good question, and we believe it is an open research question for future work. Indeed, there have been fuzzers written for general programming languages (for instance, AFL can fuzz JavaScript code for systems like V8, etc.), but it is not obvious yet how to use that appropriately for an execution guided system like ours. We think this is an exciting future direction!
> >
> > This is one point that wasn't obvious to me, since it's natural to view this as a "toy" version of general code generation, so i would suggest specifically calling this out as a limitation of the current work.

---

> > > ### Author Response · Authors · 2024-11-27
> > >
> > > Makes sense, we included a limitations section in the paper highlighting that scaling up our approach to more complex grammars will still require new ideas, in Lines 481 through 485. Thank you for your suggestion, time, and discussions!

---

### Meta-Review · Area_Chair_wPnm · 2024-12-13

**Metareview:**

This paper introduces a novel diffusion-based approach to program synthesis, leveraging syntax trees to iteratively refine programs while preserving syntactic validity. The authors position their method as an alternative to traditional autoregressive models, which are limited by sequential errors and a lack of execution feedback. By modeling program edits as a diffusion process, the approach effectively integrates syntax-aware transformations with execution-guided search, making it particularly suitable for tasks like inverse graphics. The integration of execution outputs into the synthesis pipeline further enhances the model's ability to refine programs to meet desired specifications.

The paper demonstrates strong empirical results on controlled benchmarks such as CSG2D and TinySVG, showing improvements over competitive baselines. Comprehensive ablation studies highlight the contributions of key components, including the use of tree-based diffusion and search-guided optimization. The method is presented clearly, with supplementary videos and an interactive demo that significantly aid understanding and replicability. While the scope of the current work is limited to simpler grammars, the authors acknowledge this as a future direction, and the framework’s scalability to more complex programming tasks holds significant promise. Overall, this paper offers a compelling contribution to neurosymbolic AI and program synthesis and is well-suited for acceptance.

**Additional Comments On Reviewer Discussion:**

During the rebuttal period, the authors addressed key concerns raised by reviewers, significantly improving the clarity and completeness of the paper. They revised Figure 3 and Section 3.4 to better explain syntax tree mutations and architectural details, explicitly added a limitations section discussing scalability to complex grammars, and conducted new experiments comparing their approach to recent baselines like LLaVA. Additionally, they clarified computational efficiency metrics and the rationale behind training choices involving suboptimal intermediate states. These changes, alongside thoughtful responses, demonstrated a strong commitment to improving the paper,

---

### Decision · Program_Chairs · 2025-01-22

Accept (Spotlight)